# Implementing a Feasible Exercise Programme in an Allogeneic Haematopoietic Stem Cell Transplantation Setting—Impact on Physical Activity and Fatigue

**DOI:** 10.3390/ijerph17124302

**Published:** 2020-06-16

**Authors:** Annika Kisch, Sofie Jakobsson, Anna Forsberg

**Affiliations:** 1Institute of Health Sciences, Lund University, SE-221 00 Lund, Sweden; anna.forsberg@med.lu.se; 2Department of Haematology, Skåne University Hospital, SE-222 45 Lund, Sweden; 3Institute of Health and Care Sciences, University of Gothenburg, SE-405 30 Gothenburg, Sweden; sofie.jakobsson@fhs.gu.se; 4Department of Thoracic Surgery, Skaåne University Hospital, SE-222 45 Lund, Sweden

**Keywords:** allogeneic haematopoietic stem cell transplantation, fatigue, exercise

## Abstract

Physical exercise for patients treated with allogeneic haematopoietic stem cell transplantation (allo-HSCT) has shown positive effects on the quality of life and fatigue in experimental trials. However, there is a need for longitudinal evaluation of exercise programmes implemented in a real-world clinical setting. The aim of this prospective study was to evaluate the impact of an exercise programme introduced before allo-HSCT on physical activity and fatigue before, during and after in-patient care. A structured exercise programme, including strength and endurance exercises, was implemented at a Swedish university hospital four weeks before transplantation, continuing during in-patient care and after discharge. Between March 2016 and May 2018, 67 adult patients, 33 women and 34 men with a mean age of 55.5 years participated. Fatigue was measured by the Multidimensional Fatigue Inventory at four time points. The patients documented their exercises on a checklist each week during the entire study period. The fatigue trajectory differed between various sub-groups, thus individualized supervision and support to maintain motivation is needed. In conclusion, a structured yet realistic exercise programme before admission is beneficial for allo-HSCT patients in order to reduce fatigue and prepare them for transplantation both physically and mentally.

## 1. Introduction

Allogeneic haematopoietic stem cell transplantation (allo-HSCT) is mainly a treatment for haematological malignancies, with the potential to cure the disease. In Europe more than 17,000 allo-HSCTs are performed every year, 300 of them in Sweden [1]. The treatment is very demanding with a long period of in-patient care and rehabilitation and associated with numerous side-effects and risks of various complications [1,2]. Infections, graft versus host disease (GvHD) [1], nutritional problems, mucositis, pain and negative effects on psychological and psychosocial well-being are exhausting [3,4,5]. The majority of patients are immobilized and experience high levels of fatigue due to the side-effects, the long duration of in-patient care and the burden of post-transplant complications [4,6].

Cancer-related fatigue (CRF) is defined as a “distressing, persistent, subjective sense of physical, emotional or cognitive tiredness or exhaustion related to cancer or cancer treatment that is not proportional to recent activity and interferes with usual functioning” [7]. CRF affects the majority of cancer patients and is caused by various factors; the cancer itself, the different treatments as well as psychological conditions [7,8]. CRF is common among allo-HSCT patients and has been shown to affect their health and reduce their quality of life in both the short- and the long-term, including their ability to return to work [6,9,10,11,12].

Previous research on physical exercise during cancer treatment has shown promising results with beneficial effects on fatigue and psychosocial well-being [13]. A supervised exercise programme for colon cancer patients during chemotherapy reduced physical and general fatigue [14] and physical exercise during chemotherapy had beneficial effects on health and CRF [15,16,17].

Several studies have shown promising results of physical exercise for patients treated with allo-HSCT with a positive effect on the quality of life and fatigue [13,18,19]. The physical exercise programme introduced by Wiskeman et al. [19] for allo-HSCT patients before treatment and continued during in-patient care and after discharge had significant results in terms of reducing CRF and improving quality of life, physical capacity, pain and distress. Three recently published reviews and meta-analysis [13,20,21] and one White Paper [22] highlight recommendations for exercise programmes to be incorporated into routine HSCT care starting before allo-HSCT. However, there is no universal standardized physical exercise protocol or programme for patients treated with allo-HSCT. Thus, there is a need to evaluate exercise programmes implemented in everyday practice and hence feasible in the real-world clinical setting. The aim of this study was to evaluate the impact of an exercise programme introduced before allo-HSCT on physical activity and fatigue before, during and after in-patient care.

## 2. Materials and Methods

### 2.1. Context and Setting

Before the exercise programme was implemented at the haematology clinic in a Swedish university hospital there was no structured exercise programme for allo-HSCT-patients. Instead, the patients were informed, on the day of admission and during in-patient care about the importance of not staying in bed all the time, taking short walks, stretching exercises, etc., mainly by the physiotherapist. As this was insufficiently structured, a structured but feasible exercise programme was developed and introduced for all allo-HSCT patients in March 2016 based on previous research on the benefits of physical exercise in conjunction with cancer treatment. This approach meant that exercise was not supervised or monitored by the researchers but was instead built on the patients’ motivation to be proactive and prepared for allo-HSCT.

### 2.2. The Exercise Programme

Inspired by Wiskemann et al. [19] the exercise programme started about four weeks before transplantation for all patients planned for allo-HSCT. The exercise programme contained strength and endurance exercises. Four weeks before admission the patients received verbal and written instructions from the physiotherapist including text and pictures, together with resistance bands for exercise. They were advised to exercise for 30–60 min per day, based on their own performance and health condition. The endurance exercises consisted of walking, running or cycling for 20–30 min/day. The strength exercises were tailored for the arms, shoulders, back and core using the resistance bands, calf raises, sit-ups, hip lifts and various stretching exercises, mainly for the legs. During the weeks before admission the physiotherapist or the nurse at the outpatient clinic contacted each patient by telephone on about two occasions to support her/him and sustain motivation.

The exercise programme during in-patient care also contained exercises for strength and endurance and the instructions were the same as before allo-HSCT except for running, with strength exercises for the shoulders and more focus on stretching based on the patients’ own performance and health status. During in-patient care the physiotherapist supervised the exercise programme from Monday to Friday and the rest of the staff encouraged the patients to exercise.

At discharge the patients were encouraged to continue the exercise programme they followed before the allo-HSCT, based on their own performance status and condition. The intention was for the nurse and/or the physiotherapist to continuously evaluate the patients’ efforts and support their motivation during regular follow-up at the post-HSCT outpatient clinic. However, this goal was only partly fulfilled due to a lack of staff resources at that point in time. This might have reduced the amount of exercises performed.

### 2.3. Participants and Procedure

Between 20 March 2016 and 1 May 2018 a total of 102 patients were planned for allo-HSCT. Of these, 93 were consecutively approached, informed about the study and invited to participate. Inclusion criteria: an adult (≥18 years) scheduled for allo-HSCT and able to read and write Swedish. Nine patients were not approached due to their inability to read and write Swedish. The patients received both verbal and written information about the study and gave their informed consent. Of the 93 patients seven were subsequently excluded due to cancellation of the allo-HSCT, while 14 declined participation and five dropped out for an unknown reason, resulting in a final group of 67 participants. Standard-risk patients were defined as those with acute leukaemia in first remission, chronic myelogenous leukaemia in the first chronic phase or aplastic anaemia without prior immunosuppressive therapy. All other patients were considered high-risk. Myeloablative conditioning was defined as those patients receiving cyclophosphamide and total body irradiation or fludarabine and busilvex for 4 days, while all other conditioning regimens were considered non-myeloablative (non-MAC).

Relevant clinical variables were obtained from the patients’ medical records. The multidimensional fatigue inventory (MFI-20) [23] was filled in at four different time points: four weeks before admission for allo-HSCT when the patients received the exercise programme, on admission, at discharge and three months after transplantation.

The patients were advised to make notes about their exercise every day on a checklist (one checklist per week), at all four time points during the study, what exercises they performed and for how long. However, when evaluating the checklists with their notes the level of detail was insufficient regarding type and duration of exercise. Thus, we decided to summarize whether or not they had performed endurance and/or strength exercises and for how many days and weeks.

### 2.4. The Fatigue Instrument

Fatigue was the primary outcome and measured by the MFI-20, which consists of 20 items measuring five dimensions of fatigue: general fatigue (GF), physical fatigue (PhF), reduced activity (RA), reduced motivation (RM) and mental fatigue (MF) [23]. In the questionnaire the respondents are asked to rate how they felt during the preceding days on a five-point Likert scale from ‘yes, that is accurate’ to ‘no, that is not accurate’; range 4–20 in each dimension where higher value indicates more fatigued. In line with the results of the Swedish validation of the instrument a 19-item version was used [24], where Cronbach’s α ranged from 0.67 to 0.94. Severity of fatigue was divided into three groups and estimated based on the general fatigue subscale (GF), as follows: low fatigue (score 4–11), high fatigue (score 12–15) and severe fatigue (score 16–20) [25].

### 2.5. Statistics and Ethics

SPSS 24.0 software (IBM, Chicago, IL, USA) was used to analyse the data. Descriptive statistics with median and percentiles were employed to describe the characteristics of the participants. As most data were ordinal the Wilcoxon signed ranks test was used to analyse the differences in fatigue scores between dependent groups over time and the Mann Whitney U between independent sub-groups. The significance level was set to *p* ≤ 0.05.

The study was approved by the Regional Ethical Review Board for Southern Sweden (Dnr 2016/258).

## 3. Results

The study group comprised of 67 patients, 33 women and 34 men, with a mean age of 55.5 years (SD 12.4 years, 21–70 years) the characteristics of which are presented in Table 1. The group performed strength exercises during a median of three weeks and endurance exercises for a median of four weeks before admission. The findings will be presented systematically from the four measurement points, i.e., four weeks before admission (baseline), on admission, at discharge and three months after the allo-HSCT. At each measurement point there were internal drop-outs, therefore the response rate varies; baseline *n* = 57, on admission *n* = 55, at discharge *n* = 52 and three months post allo-HSCT *n* = 50. The change between baseline and admission was significant regarding GF (*p* = 0.035), PhF (*p* ≤ 0.001) and RA (*p* = 0.005). No change was observed regarding RM (*p* = 0.501) and MF (*p* = 0.932). The fatigue trajectory for the whole group is presented in Figure 1.

During in-patient care, which lasted for a median 31 days (minimum 22 days and maximum 57 days), the patients received regular physiotherapy in accordance with the exercise programme and documented their efforts on the checklists. They performed strength exercises for a median of three weeks and endurance exercises for a median of four weeks based on their individual performance ability. At discharge the fatigue scores were significantly higher in all dimensions than on admission; GF (*p* ≤ 0.001), PhF (*p* ≤ 0.001), RA (*p* ≤ 0.001), RM (*p* = 0.034) and MF (*p* = 0.017). At discharge the scores were also higher than at baseline in terms of GF (*p* = 0.001), PhF (*p* = 0.019) and RA (*p* = 0.002). After discharge they continued exercising, but no notes about their exercises were made, and three months after discharge the fatigue scores were significantly lower in the GF (*p* = 0.012), PhF (*p* = 0.018) and RA (*p* = 0.006) dimensions compared to discharge. Three months after transplantation they had returned to the baseline level with no differences in any of the fatigue dimensions (Figure 1). As shown in Figure 2, the proportions of patients with low, high or severe fatigue differed between the different measurements points, with the highest proportion of those with severe fatigue at discharge after HSCT.

### 3.1. Sub-Group Analysis

#### 3.1.1. Gender

The mean age among the sexes was similar, women 54.5 years (SD 13.2 years, 21–69 years) and men 56.4 years (SD 11.7 years, 28–70 years). Prior to admission the male and female patients had performed a similar amount of endurance exercises, mean 3.97 weeks for the women and 3.39 weeks for the men (*p* = 0.09). However, the women had performed strength exercises for a significantly longer time, mean 3.35 weeks, than the men, mean 2.44 weeks (*p* = 0.05). The duration of the in-patient endurance exercises was equal (*p* = 0.39) for the men (mean 3.56 weeks) and women (mean 3.79 weeks), but the strength exercises differed, as the women performed them for a significantly (*p* = 0.05) longer period (mean 3.39 weeks) than the men (mean 2.65 weeks).

There were no gender differences in any fatigue dimension at any measurement point. However, when exploring the fatigue change trajectory different patterns were identified among men and women as shown in Table 2. The male patients reported significantly lower scores in the dimensions GF (*p* = 0.008), PhF (*p* = 0.001) and RA (*p* = 0.029) on admission compared to baseline. At discharge they reported significantly higher scores in all fatigue dimensions compared to admission, GF (*p* = 0.001), PhF (*p* = 0.004), RA (*p* = 0.002), RM (*p* = 0.05) and MF (*p* = 0.002). Compared to baseline there were clinical but no statistical differences in any fatigue dimension for the men, suggesting that they might already be back to baseline at discharge. Three months after the HSCT the male patients reported significantly lower fatigue compared to discharge in all dimensions except for MF; GF (*p* = 0.035), PhF (*p* = 0.032), RA (*p* = 0.008) and RM (*p* = 0.021). At this stage, they reported fatigue at the same level as at baseline with no significant differences in any dimension.

The female trajectory revealed that they reported significantly lower fatigue on admission compared to baseline in PhF (*p* = 0.024). At discharge the fatigue scores were significantly higher than on admission in three of the five dimensions, GF (*p* = 0.003), PhF (*p* ≤ 0.001) and RA (*p* ≤ 0.001). The same pattern emerged when comparing discharge with baseline levels of GF (*p* = 0.004), PhF (*p* = 0.018) and RA (*p* = 0.007). Thus, in contrast to the male patients, the female patients had not returned to their baseline levels at discharge. Even after three months the female fatigue scores were the same as the discharge scores. Compared to baseline the women reported significantly higher RA (*p* = 0.039) after three months, while their scores had returned to baseline in the other four fatigue dimensions. Thus, despite the significantly longer duration of strength exercises performed by the female patients they were more fatigued both at discharge and three months after the HSCT than the male patients, who were already back to baseline at the time of discharge.

#### 3.1.2. Age

When comparing baseline with the time of admission there were no differences between the two age groups (≤50 years; >50 years) in any fatigue dimension. However, at discharge the older patients reported significantly higher scores regarding RM (*p* = 0.018) compared to admission. When exploring the fatigue trajectory there were different patterns in the two age groups. The younger patients reported significantly lower PhF scores (*p* = 0.024) at/on admission than at baseline. Regarding the other four dimensions there were clinical but not significant differences, as shown in Table 3. At discharge they reported significantly higher PhF (*p* = 0.015) and RA (*p* = 0.032) but statistically equal to baseline levels in the other fatigue dimensions. Three months after the HSCT they reported lower scores in PhF (*p* = 0.037) compared to discharge, with no statistical differences in the other four dimensions. Compared to baseline PhF and RA had clinically improved after three months, but there were no significant differences in any of the five fatigue dimensions. The older patients reported significantly lower scores in GF (*p* = 0.040), PhF (*p* = 0.001) and RA (*p* = 0.015) on admission compared to baseline. At discharge the older patients reported higher scores than on admission in all dimensions; GF (*p* ≤ 0.001), PhF (*p* ≤ 0.001), RA (*p* ≤ 0.001), RM (*p* = 0.018) and MF (*p* = 0.013). Compared to baseline, their scores were clinically higher in all dimensions at discharge, but GF (*p* = 0.013) and RA (*p* = 0.001) were significantly higher. After three months they reported significantly lower scores compared to discharge regarding GF (*p* = 0.041), RA (*p* = 0.019) and RM (*p* = 0.020). Their activity was continuously reduced (*p* = 0.048) compared to baseline with clinical but no statistical differences in the other four dimensions. Thus, the younger patients had already returned to baseline at discharge, while the older patients were persistently fatigued three months after transplantation.

#### 3.1.3. Disease Groups

As described in the Methods section, the whole group was divided into standard-risk versus high-risk. There were no differences in any fatigue dimension at any measurement point between those with a standard risk and those with a high risk. However, in this sub-group analysis the fatigue trajectory revealed different patterns. Those with a standard risk reported significantly lower scores on admission compared to baseline in GF (*p* = 0.018), PhF (*p* = 0.007) and RA (*p* = 0.044). At discharge they were again more fatigued in the same dimensions compared to admission, GF (*p* = 0.001), PhF (*p* = 0.002) and RA (*p* = 0.003). At discharge they reported persistently high GF compared to baseline (*p* = 0.009), but were statistically back to baseline in the other four dimensions despite clinical differences. Furthermore, three months after HSCT they reported lower scores in all dimensions compared to discharge and were clinically and statistically back at baseline levels in all dimensions. Those with an estimated high risk reported lower scores regarding PhF (*p* = 0.003) and RA (*p* = 0.035) on admission compared to baseline but significantly higher scores at discharge compared to admission, GF (*p* = 0.002), PhF (*p* = 0.001) and RA (*p* ≤ 0.001) as well as compared to baseline GF (*p* = 0.042), PhF (*p* = 0.045) and RA (*p* = 0.012; Table 4). After three months they were as fatigued as at discharge with no significant differences in any of the five dimensions, although being clinically better in all dimensions except MF. Compared to baseline they scored significantly higher in RA (*p* = 0.013), RM (*p* = 0.05) and MF (*p* = 0.001). Thus, the high-risk group was more longitudinally affected by fatigue, especially the mental aspects, than the low-risk group.

#### 3.1.4. Conditioning Regimens

The patients received different conditioning regimens, either myeloablative [2] or non-myeloablative (non-MAC). The mean age in the MAC group was 37.5 years (SD 10.5 years, 21–52 years) and in the non-MAC group 59.8 years (SD 8.3 years, 27–70 years; *p* ≤ 0.001). The MAC group reported a significantly higher MF score at baseline (*p* = 0.014) than the non-MAC group with clinical but no statistical differences in any other dimension at any measurement point (Table 5).

The non-MAC patients reported significantly lower scores on admission compared to baseline in GF (*p* = 0.05), PhF (*p* = 0.001) and RA (*p* = 0.028) but significantly higher scores in four of five dimensions at discharge compared to admission, i.e., GF (*p* ≤ 0.001), PhF (*p* ≤ 0.001), RA (*p* ≤ 0.001) and MF (*p* = 0.028) and compared to baseline in GF (*p* = 0.016), PhF (*p* = 0.046) and RA (*p* = 0.004). Three months after HSCT, the non-MAC group reported a significantly lower GF score (*p* = 0.033) compared to discharge, but no significant differences in the other dimensions. Compared to baseline they were persistently more reduced in terms of activity (*p* = 0.033) and more mentally fatigued (*p* = 0.015). The MAC group reported lower scores at/on admission compared to baseline, PhF (*p* = 0.011) and RA (*p* = 0.014). At discharge they reported significantly higher scores, i.e., after transplantation, in GF (*p* = 0.020), PhF (*p* = 0.007) and RA (*p* = 0.018) compared to admission. Compared to baseline they were statistically but not clinically back to baseline in all dimensions except for GF with a median of 14 (*p* = 0.028). Three months after allo-HSCT, the MAC group was less fatigued regarding PhF (*p* = 0.007) and RA (*p* = 0.028) compared to discharge, but statistically equally fatigued in the other dimensions and back to baseline in all fatigue dimensions with clinically lower scores in PhF, RA and MF.

## 4. Discussion

The aim of the study was to evaluate the impact of an exercise programme introduced before allo-HSCT on physical activity and fatigue before, during and after in-patient care. To summarize the findings on the impact of fatigue, there were few or no differences in the five fatigue dimensions between the sub-groups at each of the measurement points. However, different patterns emerged when exploring the longitudinal trajectory of each sub-group resulting in the following key findings:Despite the significantly longer duration of strength exercises performed by the female patients they were more fatigued both at discharge and three months after the HSCT than the male patients, who were already back to baseline at the time of discharge.The younger patients (≤50 years) had already returned to baseline fatigue levels at discharge, while the older patients were persistently fatigued three months after transplantation.The high-risk group was more longitudinally affected by fatigue, especially the mental aspects, than the standard-risk group.There were no statistical differences in fatigue levels at any measurement point after allo-HSCT between the MAC patients and the non-MAC patients. Three months after HSCT, the non-MAC group reported a significantly lower GF score compared to discharge and the MAC group was less fatigued regarding PhF and RA compared to discharge.

Inspired by Wiskemann et al. [19] we developed and introduced a structured exercise programme for allo-HSCT patients about four weeks before admission, with the aim of reporting its impact on physical activity and fatigue before, during and after in-patient care. The majority of the patients appreciated the exercise programme; it was easy to understand and follow and it was something they could do themselves to promote their treatment and rehabilitation. The intention was that support should be received along with the exercise programme. However, due to a lack of resources there was a lack of supervision and support and some patients expressed that it was difficult to continue training without adequate support. The patients seemed to benefit from the exercise programme before admission as evidenced by reduced levels of fatigue on admission, with some gender differences. The patients showed reduced levels of fatigue over time. This might be explained by the exercise programme introduced some weeks before admission, which is consistent with results from previous studies in cancer patients [14,15,16,17] and patients treated with allo-HSCT [18,19]. The newly published reviews and recommendations [13,20,21,22] point out that there is no universal standardized exercise programme or protocol for allo-HSCT patients.

### 4.1. Considerations Regarding Age Differences

We expected the patients to be more fatigued after allo-HSCT than before, according to earlier studies [6,11,12] and our clinical experience. We also expected the older patients to be more fatigued than the younger patients, which was evident in the findings. As expected, and not surprising, younger patients (≤50 years) recovered in fatigue levels earlier than the older ones. This means that older patients often need more support. A reasonable implication is that their exercise programme should be prolonged and adapted in its content, while the younger patients are expected to be able to cope on their own earlier. Furthermore, the older patients seemed to be more negatively affected by the treatment than the younger ones, which is also linked to the result that the patients who received non-myeloablative conditioning (non-MAC) were older than the ones receiving MAC and they still reported reduced activity and more mental fatigue three months after transplantation compared to baseline.

### 4.2. Reflections on Gender Differences

We expected gender differences since previously it has been described that women report more fatigue than men after allo-HSCT [11]. However, the results of previous studies concerning gender differences are not consistent [6,11,12]. Since women in general report more symptoms than men it is possible that they also are more burdened by fatigue [9]. The fact that female patients were more fatigued both at discharge and three months after the HSCT than the male patients was somewhat surprising since they were more diligent in their exercise than the male patients. The reason why female patients needed longer time for recovery concerning fatigue might be physiological. However, we argue that an additional explanation might be that women often take more domestic responsibility; i.e., for the duties at home and their family and therefore are more fatigued during a longer follow-up time. Thus, we suggest that an occupational therapist is enrolled performing an activity analysis and then provide tools for activity balance during the recovery period, specifically for the female allo-HSCT recipients.

### 4.3. Disease Groups

We expected to find some differences in fatigue levels between high-risk patients and standard-risk patients. Hardly surprising the high-risk patients were more longitudinally affected by fatigue compared to the standard-risk patients, with the possible explanation that their diseases are more aggressive and they have received more demanding treatment before allo-HSCT. The patients with an estimated standard risk were already at discharge almost back to baseline, while those in the estimated high-risk group were still continuously mentally fatigued after three months.

### 4.4. Conditioning Regimens

We also expected the MAC patients to be more fatigued than the non-MAC group, since the MAC conditioning is more demanding than the non-MAC. However we were surprised by the magnitude of age difference between the two groups. There were no statistical differences in fatigue between the non-MAC group and the MAC group at the follow-up measurements points. Three months after allo-HSCT both groups were less fatigued compared to discharge. The lack of differences most presumably is linked to the age differences, mean age 37.5 years vs. 59.8 years, and therefore the MAC patients are able to recover faster.

Based on the sub-group analysis our results suggest that it is difficult to implement one standard exercise programme due to the fact that the fatigue trajectories differ depending on age, gender, disease status, conditioning regimen and individual differences. However, we can confirm that exercise training before admission for allo-HSCT is beneficial for patients in terms of reducing fatigue. It probably also makes the patients more prepared both physically and mentally. Therefore, an individualized, supervised exercise programme should be mandatory before, during and after allo-HSCT and is possible to implement even with limited resources.

### 4.5. Strengths and Limitations

The strengths of this study are the participation rate of 72% and the fact that it was performed as a real-world study with a longitudinal design. The internal drop-outs along the treatment trajectory is inevitable in this patient group due to the severity of the diseases and the treatment. However, a limitation is that we do not have exact number of reasons for drop-outs. The majority of the patients were positive about the exercise programme and happy to receive something they could do themselves to promote their treatment and rehabilitation. The exercise programme was appreciated by the participants because it was easy to understand and follow as it contained short written information complemented by illustrative photos. The limitations are mainly due to shortcomings in everyday clinical practice and the high staff turnover in university hospitals. When the exercise programme was introduced the intention was that the patients should receive support along the treatment trajectory. However, there was a lack of consistent supervision due to reorganization of the tasks of the physiotherapists at the clinic. Some patients expressed that it was difficult to continue training without adequate support. However, the transplant nursing staff tried to be as active as possible in encouraging the patients. Due to the organizational shortcomings the level of details in the checklists was insufficient and information about the type and daily duration of exercise was missing. Thus, these aspects could not be analysed as intended. However, information was obtained about whether the patients performed endurance and/or strength exercises and for how many days and weeks, which enabled some relevant analyses and conclusions. The lack of staff resources resulting in limited possibilities for evaluation and support during follow-up illustrates the clinical conditions in the real world health care. This might have reduced the amount of exercises performed, which in turn may have resulted in higher levels of fatigue.

## 5. Implications and Future Research

Our study confirms earlier results that individualized supported exercise training for patients treated with allo-HSCT should be incorporated into routine care and start some weeks before admission, in order to reduce fatigue and improve patients’ well-being. Individual exercise capacity should be evaluated before, during and after allo-HSCT, in order to guide the staff in their supportive work. According to our results additional focus on exercise should be given to female patients, older and high-risk patients. To involve an occupational therapist performing activity assessment and promoting activity balance might be highly useful. However, further studies are needed to explore the impact of support and encouragement during exercising on levels of fatigue, quality of life and well-being.

## 6. Conclusions

In conclusion, a structured yet realistic exercise programme is suitable and beneficial before admission for allo-HSCT in order to reduce fatigue and prepare the patients for the transplantation both physically and mentally. Our findings confirmed previous research and recommendations [13,20,21,22] on the benefit of beginning exercise training before transplantation. The fatigue trajectory differs between the various sub-groups and thus person centred support to maintain motivation is necessary for up to at least three months after transplantation.

## Figures and Tables

**Figure 1 ijerph-17-04302-f001:**
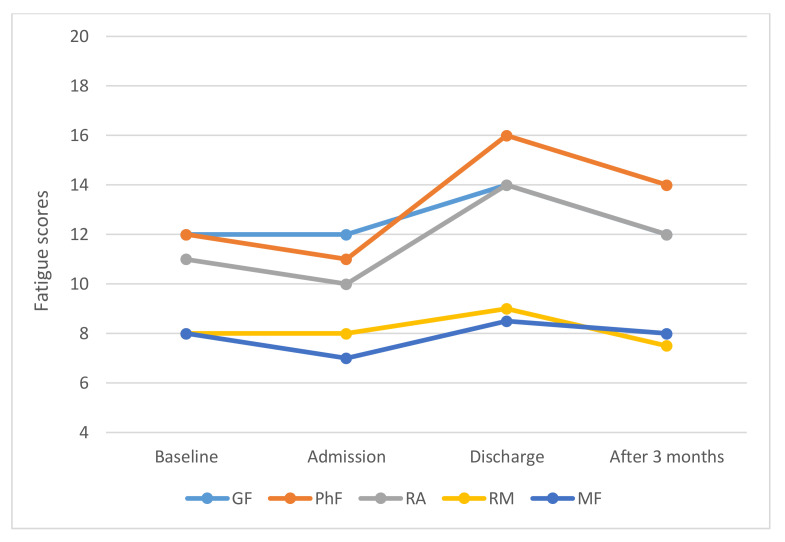
The fatigue trajectory of the whole group of allogeneic haematopoietic stem cell transplantation (allo-HSCT) patients shown by the median levels. GF, General Fatigue; PhF, Physical Fatigue; RA, Reduced Activity; RM, Reduced Motivation and MF, Mental Fatigue (range 4–20).

**Figure 2 ijerph-17-04302-f002:**
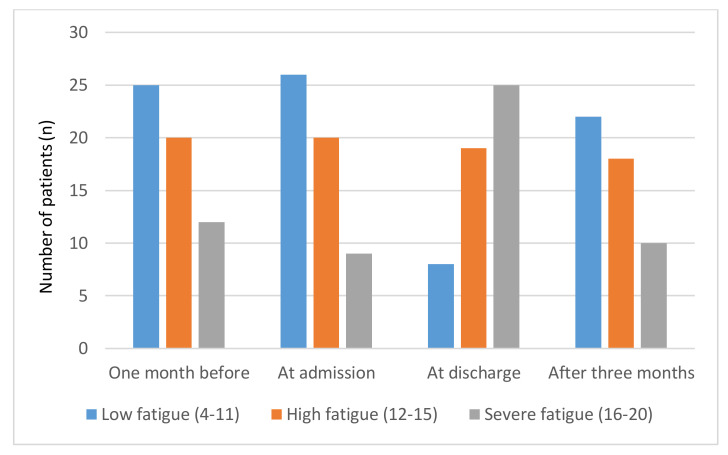
Proportions of the number of patients with low, high or severe fatigue at the different measurements points.

**Table 1 ijerph-17-04302-t001:** Patient characteristics (*n* = 67).

Characteristics	*N* (%)
*Age, years; mean (SD)* *range* ≤50 years >50 years	*55.5 (12.4)* *21–70* 17 (25) 50 (75)
*Gender, n (%)*	
male	34 (51)
female	33 (49)
*Primary diagnosis*	
AML	25 (37)
MDS	14 (21)
MPS	9 (13)
NHL	8 (12)
ALL	6 (9)
CML	4 (6)
CLL	1 (2)
*Disease status at transplant*	
standard-risk	28 (42)
high-risk	39 (58)
*Donor*	
sibling	16 (24)
URD	47 (70)
haplo	4 (6)
*Stem cell source*	
BM	0 (0)
PBSC	67 (100)
*Conditioning regimen*	
MAC	13 (19)
Non-MAC	54 (81)
*Marital status at transplant*	
Married/cohabiting	48 (72)
Living alone	19 (28)

Abbreviations: ALL, acute lymphatic leukaemia; AML, acute myeloid leukaemia; BM, bone marrow; CLL, chronic lymphatic leukaemia; CML, chronic myeloid leukaemia; haplo, haplo-identical donor; MAC, myeloablative conditioning; MDS, myeloid dysplastic syndrome; MPS, myeloproliferative syndrome; NHL, non-Hodgkin lymphoma; PBSC, peripheral blood stem cells; non-MAC, non-myeloablative conditioning; URD, unrelated donor.

**Table 2 ijerph-17-04302-t002:** The trajectory of fatigue levels among men and women undergoing allo-HSCT.

Fatigue Dimensions	Baseline: One Month Before Admission Median (p_25_; p_75_)		On Admission For HSCT Median (p_25_; p_75_)		At Discharge After HSCT Median (p_25_; p_75_)		3 Months After HSCT Median (p_25_; p_75_)	
Gender	Men (*n* = 29)	Women (*n* = 28)	Men (*n* = 28)	Women (*n* = 27)	Men (*n* = 25)	Women (*n* = 27)	Men (*n* = 26)	Women (*n* = 24)
General fatigue (GF)	12 (7; 16)	13 (8.25; 15)	10.5 (7.25; 14.5)	12 (9; 14)	13 (12; 16.5)	14 (12; 18)	12 (10.75; 14.25)	13 (8.25; 15.75)
Physical fatigue (PhF)	12 (8; 18.5)	13 (10; 17)	11 (7; 14.5)	12 (10; 14)	17 (12.5; 18.5)	15 (13; 18)	13 (10.75; 16)	15 (10.25; 17)
Reduced activity (RA)	11 (8; 14)	11.5 (8.25; 15)	9.5 (7; 14.75)	10 (8; 13)	14 (10.5; 16)	14 (12; 15)	11.5 (8; 14.25)	13 (7; 16)
Reduced motivation (RM)	8 (6; 10.5)	8.5 (6; 12)	7.5 (5; 10.75)	9 (7; 14)	10 (7.5; 12)	9 (6; 13)	7 (5; 10.25)	8 (5; 13.5)
Mental fatigue (MF)	8 (4; 12)	8 (4; 9.75)	7.5 (4.25; 10.5)	7 (4; 12)	11 (7.5; 11.5)	7 (4; 11)	8 (4; 11)	8 (4; 12)

Due to internal drop-outs: baseline *n* = 57, on admission *n* = 55, at discharge *n* = 52 and three months post allo-HSCT *n* = 50.

**Table 3 ijerph-17-04302-t003:** The trajectory of fatigue levels related to age among patients undergoing allo-HSCT.

Fatigue Dimensions	Baseline: One Month Before Admission Median (p_25_; p_75_)		On Admission For HSCT Median (p_25_; p_75_)		At Discharge After HSCT Median (p_25_; p_75_)		3 Months After HSCT Median (p_25_; p_75_)	
Age group	≤50 years (*n* = 16)	>50 years (*n* = 41)	≤50 years (*n* = 14)	>50 years (*n* = 41)	≤50 years (*n* = 15)	>50 years (*n* = 37)	≤50 years (*n* = 13)	>50 years (*n* = 37)
General fatigue (GF)	13.5 (9.75; 15)	11 (7.5; 15)	12 (8.75; 16)	11 (8; 13.5)	14 (13; 18)	14 (12; 17.5)	13 (11; 15.5)	12 (8.5; 15)
Physical fatigue (PhF)	14 (10.25; 17)	12 (10; 18)	11.5 (9; 15.25)	11 (9; 14)	16 (13; 18)	16 (13; 18)	12 (10.5; 16)	14 (10.5; 17)
Reduced activity (RA)	13 (9.25; 15)	11 (8; 13.5)	9.5 (7; 14.25)	10 (6.5; 14)	13 (11; 15)	14 (12; 17)	10 (7; 15.5)	12 (8; 15)
Reduced motivation (RM)	8.5 (6; 11.5)	8 (6; 12)	7.5 (4.75; 11)	9 (5.5; 12.5)	7 (6; 10)	11 (8; 13)	8 (5; 11.5)	7 (5; 11.5)
Mental fatigue (MF)	9.5 (7; 12)	5 (4; 10)	8.5 (6.25; 12.25)	7 (4; 10)	9 (7; 11)	8 (4; 12)	9 (7; 12)	8 (4; 11)

Due to internal drop-outs: baseline *n* = 57, on admission *n* = 55, at discharge *n* = 52 and three months post allo-HSCT *n* = 50.

**Table 4 ijerph-17-04302-t004:** The trajectory of fatigue levels related to disease status among patients undergoing allo-HSCT.

Fatigue Dimensions	Baseline: One Month Before Admission Median (p_25_; p_75_)		On Admission For HSCT Median (p_25_; p_75_)		At Discharge After HSCT Median (p_25_; p_75_)		3 Months After HSCT Median (p_25_; p_75_)	
Disease group	standard-risk (*n* = 25)	high-risk (*n* = 32)	standard-risk (*n* = 24)	high-risk (*n* = 31)	standard-risk (*n* = 23)	high-risk (*n* = 29)	standard-risk (*n* = 23)	high-risk (*n* = 27)
General fatigue (GF)	13 (8; 15.5)	12 (8; 14.75)	11.5 (8; 14.75)	12 (8; 14)	14 (12; 18)	14 (12; 16.5)	12 (10; 15)	13 (9; 15)
Physical fatigue (PhF)	12 (10; 17.5)	12.5 (10; 17.5)	11 (9; 13.75)	11 (10; 15)	16 (13; 17)	16 (13.5; 18)	13 (10; 17)	15 (12; 16)
Reduced activity (RA)	12 (6; 15.5)	11 (9; 13)	10 (6.25; 15)	10 (8; 14)	14 (11; 16)	14 (12; 16)	12 (7; 15)	12 (8; 15)
Reduced motivation (RM)	8 (6; 11.5)	8 (6; 12)	8 (5; 12.75)	9 (6; 12)	10 (6; 13)	9 (6.5; 12)	7 (5; 12)	8 (5; 11)
Mental fatigue (MF)	10 (4; 12.5)	6 (4; 8.75)	8 (4.25; 11.75)	7 (4; 11)	9 (7; 11)	8 (4; 12)	7 (4; 11)	9 (4; 12)

Due to internal drop-outs: baseline *n* = 57, on admission *n* = 55, at discharge *n* = 52 and three months post allo-HSCT *n* = 50.

**Table 5 ijerph-17-04302-t005:** The trajectory of fatigue levels related to conditioning regimens among patients undergoing allo-HSCT.

Fatigue Dimensions	Baseline: One Month Before Admission Median (p_25_; p_75_)		On Admission For HSCT Median (p_25_; p_75_)		At Discharge After HSCT Median (p_25_; p_75_)		3 Months After HSCT Median (p_25_; p_75_)	
Conditioning regimen	non-Mac (*n* = 45)	Mac (*n* = 12)	non-Mac (*n* = 44)	Mac (*n* = 11)	non-Mac (*n* = 41)	Mac (*n* = 11)	non-Mac (*n* = 39)	Mac (*n* = 11)
General fatigue (GF)	12 (8; 15)	12.5 (6; 15.75)	11.5 (8, 13.75)	12 (8; 16)	14 (12; 16.5)	14 (13; 19)	12 (9; 15)	15 (11; 20)
Physical fatigue (PhF)	12 (10; 17)	16.5 (7.75,17.75)	11 (9; 14)	12 (9; 16)	15 (12.5; 17.5)	17 (13; 20)	14 (11; 17)	15 (10; 16)
Reduced activity (RA)	11 (8; 13.5)	13 (7.5,15.7)	10 (7.25; 14)	9 (7; 15)	14 (12; 16)	14 (11; 16)	12 (8; 15)	10 (7; 16)
Reduced motivation (RM)	8 (6; 12)	7.5 (6, 11.25)	9 (6; 12.75)	6 (4; 9)	10 (6.5; 13)	9 (6; 11)	7 (5; 11)	8 (5, 12)
Mental fatigue (MF)	7 (4; 9)	11 (9, 12.75)	7 (4; 9)	9 (4; 13)	8 (4; 11.5)	11 (7; 11)	8 (4; 11)	9 (7; 12)

Due to internal drop-outs: baseline *n* = 57, on admission *n* = 55, at discharge *n* = 52 and three months post allo-HSCT *n* = 50.

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
