# Peer review of "Implementing a Feasible Exercise Programme in an Allogeneic Haematopoietic Stem Cell Transplantation Setting—Impact on Physical Activity and Fatigue"

_ijerph, 2020, doi:10.3390/ijerph17124302_

Round 1
Reviewer 1 Report
The fatigue instrument: please add reliability coefficients.
Statistics: Please add effect sizes.
Results: I recommend changing Tables 2 – 5 to figures with standard error bars.
Discussion: The Discussion should contain most of the following elements:
- Restate the purpose and summarize the results
- Restate each aim (or hypothesis if any) and discuss how the data fulfilled that aim and if the results agree or do not agree with previously cited literature in the Introduction
- Discuss theoretical and practical implications of the results (if possible)
- Identify study strengths and limitations
- Provide recommendations for future research
- Summarize and state conclusions
Please check for these items in the Discussion.
Author Response
Thank you very much for reviewing our manuscript and giving your suggestions and recommendations which certainly have improved our manuscript.
- We have added the reliability coefficients for the fatigue instrument MFI-20, line 126
- No effect size measures were calculated since the design was not a RCT or non-randomized experimental trial.
- We appreciate your recommendation to change tables 2 – 5 to figurers with standard error bars. We also agree that it probably also would make the result presentation more appealing to the readers. However, since we have chosen to consider all variables as ordinal and use IQR instead of mean, SD and confidence intervals we can’t transform the tables to recommended figures. In line with no comments about the tables from the two other reviewers we choose to keep the tables as they are.
- We have carefully reviewed your suggestions for items to consider in the Discussion and several revisions according to these have been made in the Discussion, line 285 – 386.
Reviewer 2 Report
Q1:ABSTRACT
A large part of the RESULTS section showed the subgroup analysis results. If subgroup analysis has practical significance, it is suggested to show it in the ABSTRACT.
Q2:MATERIALS AND METHODS
As shown in line 94-95 “However, this goal was only partly fulfilled due to a lack of staff resources at that point in time.” Does this have a potential impact on the data analysis?
Q3:DISCUSSION
The first paragraph of the DISCUSSION seems more like a detailed description of the results than a demonstration of the core results. It is suggested that this paragraph be revised to highlight the discussion of core results.
The second paragraph of the DISCUSSION is not enough. The in-depth discussion should be added, and possible explanations should be given to the research results.
Q4:REFERENCES
The REFERENCES needs to be updated. It is better to quote the reference of the past five years.
Author Response
We deeply appreciate your review enabling us to revise and improve our manuscript according to your excellent suggestions.
AR= Author’s response
- Q1: ABSTRACT ’If subgroup analysis has practical significance, it is suggested to show it in the ABSTRACT.’ AR: We believe that the subgroup analysis has practical influence, and the most important clinical implication is to individualize the support and supervision. Therefore we chose to keep our original sentence in the Abstract as follows: ’thus individualized supervision and support to maintain motivation is needed’, line 24-25
- Q2: MATERIALS AND METHODS, Line 94-95, ‘… due to lack of staff resources at that point of time. Does this have a potential impact on the data analysis?’ AR: We agree that this should be commented in the manuscript, and we have done so by adding: ‘This might have reduced the amount of exercises performed.’, line 94-95 and we have now also discussed this in 4.1. Strengths & Limitations section, line 383-386.
- Q3: DISCUSSION, AR: We totally agree that the Discussion needed to be revised and improved. In line also with the other reviewers, we have now made several revisions and as you suggested we have highlighted the discussion of the core results and added possible explanations to our results, line 285 – 386.
- Q4: REFERENCES. AR: We appreciate your remark regarding the references and we have now updated the references to more recently published articles, line 417 – 495.
Reviewer 3 Report
The text is very well written. The background of the research presented in the introduction as well as the description of materials and methods is on a high scientific level. The subject matter is difficult, but very important in society. My comments relate mainly to two issues: the way the results are presented and the discussion. I believe that Table 1 should present the age ranges of respondents. The information about how many people were under 50 years old and how many above is not legible. The presentation of the other results is generally correct and makes the results easier to understand. The discussion is rather weak. In line 307 the authors write that "it is difficult to compare the results of different studies as the populations and settings differ". In my opinion, this does not exempt from the discourse, but only makes it more interesting. I would encourage a broader confrontation of results. My last little remark concerns the keywords. I believe that the keywords should not be a duplication of the words used in the title of the manuscript. I hope that my comments will be useful to the authors and will make the text well received by scientists
Author Response
We deeply appreciate your encouraging comments and your support in order to improve the paper.
- The age ranges and number of patients in each age group have been clarified in Table 1, line 148
- We agree that the Discussion previously was weak.Therefore, now several revisions have been made, line 285 – 386.
- We have modified the keywords according to your suggestion, line 28